# A Scoping Review of the Risk Factors Associated with Anaemia among Children Under Five Years in Sub-Saharan African Countries

**DOI:** 10.3390/ijerph17238829

**Published:** 2020-11-27

**Authors:** Phillips Edomwonyi Obasohan, Stephen J. Walters, Richard Jacques, Khaled Khatab

**Affiliations:** 1School of Health and Related Research (ScHARR), University of Sheffield, Sheffield S1 4DA, UK; s.j.walters@sheffield.ac.uk (S.J.W.); r.jacques@sheffield.ac.uk (R.J.); 2Department of Liberal Studies, College of Administrative and Business Studies, Niger State Polytechnic, Bida Campus, Bida 912231, Nigeria; 3Faculty of Health and Wellbeing, Sheffield Hallam University, Sheffield S10 2BP, UK; K.Khatab@shu.ac.uk

**Keywords:** anaemia, iron-deficiency, under five, sub-Saharan Africa, risk factors, scoping review

## Abstract

*Background/Purpose:* Globally, anaemia is a severe public health condition affecting over 24% of the world’s population. Children under five years old and pregnant women are the most vulnerable to this disease. This scoping review aimed to evaluate studies that used classical statistical regression methods on nationally representative health survey data to identify the individual socioeconomic, demographic and contextual risk factors associated with developing anaemia among children under five years of age in sub-Saharan Africa (SSA). *Methods/Design:* The reporting pattern followed the Preferred Reporting Items for Systematic Reviews and Meta-Analyses extension for Scoping Reviews (PRISMA-ScR) guidelines. The following databases were searched: MEDLINE, EMBASE (OVID platform), Web of Science, PUBMED, Cumulative Index to Nursing and Allied Health Literature (CINAHL), PsycINFO, Scopus, Cochrane library, African Journal of online (AJOL), Google Scholar and Measure DHS. *Results:* The review identified 20 relevant studies and the risk factors for anaemia were classified as child-related, parental/household-related and community- or area-related factors. The risk factors for anaemia identified included age, birth order, sex, comorbidities (such as fever, diarrhoea and acute respiratory infection), malnutrition or stunting, maternal education, maternal age, mother’s anaemia status, household wealth and place of residence. *Conclusion:* The outcome of this review is of significant value for health policy and planners to enable them to make informed decision that will correct any imbalances in anaemia across socioeconomic, demographic and contextual characteristics, with the view of making efficient distributions of health interventions.

## 1. Introduction

Globally, anaemia is a severe public health condition affecting over 24% of the world’s population [1]. People across different regions, ages and sexes are affected by this health burden [2]. It indicates that the prevalence of anaemia cuts across developed and developing countries, males and females, children and adults. Nevertheless, children under five years of age and pregnant women are the ones who are most likely to be affected by this disease condition. Developing countries have a four times higher burden of anaemia than in developed countries [3]. Between 1990 and 2014, during the global Millennium Development Goals (MDG) watch, under-five anaemia witnessed a global decline from a prevalence of 51.4% to 41.4%, but in 2016 it gradually increased to 41.7% [4]. In Nigeria, the post MDG prevalence of under-5 anaemia has risen from 60% in 2015 to 68.1% in 2018 [5,6]. Belachew and Tewabe classified anaemia as a widespread hematologic disorder in children [7]. Diagnosis of anaemia is through blood examination for the haemoglobin or haematocrit concentration of a standard threshold by age and sex [8]. At the population level, a survey reported that using a haemoglobin (Hb) concentration test is more reliable for detecting anaemia than clinical measurement [8]. However, the same study cautioned that the mean Hb concentration level could be found to be lower in a population with high rates of inherited haemoglobinopathies [8].

The causes of anaemia, especially in developing countries, are primarily attributed to iron deficiency [9] in conjunction with other predisposing factors, such as acute and chronic infections like malaria, tuberculosis, cancer and HIV. Other causes are malnutrition and haemoglobinopathies [8]. Globally, more than 50% of cases of anaemia are caused by iron deficiency [3], and specifically over 42% of all under-5 anaemia is attributable to iron deficiency [10]. The harmful effects of under-five anaemia include long-term cognitive disorder, impaired educational performance [3,11], retardation in physical growth, poor motor skills, impaired language development [3] and an increased risk of child mortality as a result of severe anaemia [8]. Furthermore, of the relevant literature identified, other risk factors associated with under-five anaemia categorised as child-related are age, birth order, sex, nutritional status and the use of insecticide-treated nets for children [10,11,12,13]; others include household-related characteristics: wealth status, parental or caregiver’s educational attainment, employment status, age at first marriage and place of residence [10,12].

To effectively tackle the harm caused by anaemia through careful health interventions, knowing the estimates of its prevalence, associated causes and risk factors are very crucial [3]. This scoping review study is an integral part of a more extensive doctoral research work focusing on multi-morbidities among children of under five years of age in Nigeria. The relative lack of studies on anaemia among children of under five years (otherwise referred to as “under-five anaemia”) in Nigeria using nationally representative surveys has necessitated expanding the review to cover sub-Saharan Africa (SSA). Given the rising trend of under-five anaemia, identification and description of various risk factors, as well as using classical statistical regression methods and an array of different survey types, will be useful tools to enhance future health science investigations [14]. In Nigeria and perhaps in SSA, studies on scoping reviews to evaluate the risk factors that are associated with under-five anaemia are urgently needed to help programme planners and policymakers in the efficient distribution of scarce health interventions. The methodological approach of a qualitative review of many studies’ contents has made this study lean towards a scoping rather than systematic review [14]. The overall research question for this scoping review is what are the risk factors associated with the development of anaemia in children under five years of age in sub-Saharan Africa?

### The Aim of the Scoping Review

This scoping review aimed to identify and evaluate studies that performed classical regression analysis—a regression analysis that is based on Frequentist statistical methods on nationally representative health survey data to identify the individual socioeconomic, demographic and contextual risk factors associated with developing anaemia among children under five years of age in sub-Saharan Africa (SSA).

## 2. Materials and Methods

The pattern of reporting this scoping review followed the “Preferred Reporting Items for Systematic Reviews and Meta-Analyses extension for Scoping Reviews (PRISMA-ScR) guidelines” [15,16]. The scoping study also adopted the steps described in the four-phase flow chart recommended by the PRISMA statement (see Figure 1).

### 2.1. Protocol and Registration Declaration

There was a protocol study prepared for this review but it was neither registered nor published.

### 2.2. Eligibility Criteria

Studies included in the review followed the PICOTS (Population, Interventions, Comparators, Outcomes, Timing and Setting/Study Design) principles enumerated below:

#### 2.2.1. Population

Participating studies were searched for those that were conducted in countries in SSA, which addressed the individual socioeconomic, demographic and community risk factors that are related to anaemia among children under five years of age, using national health representative surveys. The studies also included both male and female children less than five years of age and who resided in any of the sub-Saharan Africa countries. We also considered for inclusion studies that involved both adults (and/or children above five years) and children under 5 years of age, taking into consideration that the data for the under-fives were reported differently from others [8].

#### 2.2.2. Interventions

Eligible for inclusion were studies that focused on predictors or risk factors or determinants of anaemia among under-five or pre-school children in SSA. In view of this, the following were considered: (i) child-related variables; (ii) parental/caregiver-related variables; (iii) household-related variables and socioeconomic status; (iv) demographic status; and (v) area characteristics. This scoping review spanned both individual and contextual exposures.

#### 2.2.3. Comparators

Given the broad dimension of the exposures of interest, we included studies involving some perspectives of comparisons. These studies suggested two mutually exclusive groups: those that are “anaemic” and those that are “not anaemic”, for which we compared the exposures. However, studies that classified anaemia status into severe, moderate and mild were included and considered as anaemic. However, we excluded studies that reported determinants of anaemia status in children by specific characteristics, such as age, maternal education and status, etc., rather than by their anaemia status.

#### 2.2.4. Outcomes

The outcome of interest for inclusion was the under-five anaemia status and was determined by the haemoglobin (Hb) level being less than 11.0 grams per decilitre. Anaemia status was, however, further classified as mild when Hb = 10–10.9 g/dL, moderate when Hb = 7.0–9.9 g/dL and severe anaemia when Hb < 7 g/dL [7]. Studies were also included that considered haemoglobin deficiency, malaria-induced anaemia, nutritional-deficiency anaemia and iron-deficiency anaemia (IDA) as the outcomes of interest. The most common contributory factor to the development of anaemia is iron deficiency and is responsible for more than 50% of anaemia cases worldwide. So, IDA and anaemia are often taken as the same [8].

#### 2.2.5. Timing or Time Frames

The studies considered for inclusion were those published between 1 January 1990 and 26 June 2020.

#### 2.2.6. Study Designs

This scoping review included observational studies, such as case-control and nested case-control studies, cross-sectional and cohort studies (prospective and retrospective) and randomised control trials, which controlled the risk factors under consideration. The search also included those studies that applied classical statistical regression methods for the analyses.

### 2.3. Information Sources

The strategy for this literature search was carried out by the candidate (P.E.O.) of the School of Health and Related Research (ScHARR), the University of Sheffield, United Kingdom. The selection of the literature for screening was based on keywords and subject headings. The following databases and grey literature, using the identified search terms, were used: MEDLINE, EMBASE (OVID platform), Web of Science, PUBMED, Cumulative Index to Nursing and Allied Health Literature (CINAHL), African Journal of online (AJOL), Google Scholar and Scopus. The search was supplemented by searching for relevant literature in the “Unishef” library, WhiteRose Research Online and Measure DHS. A hand search for potential papers from the reference list of identified documents was also conducted. Only articles written in the English language, and with a publication date between 1 January 1990 and 26 June 2020, were included.

### 2.4. Search Strategy

The literature search included searching for each key term or text words individually. The phrases were first searched in PubMed headings using the appropriate truncation and wildcard parameters [17]. The search terms applied in the PICOTS (Population, Intervention, Comparators, Outcomes, Timing and Settings) categories were as follows: demographic health survey, AIDS indicator survey, malaria indicator survey, multiple indicator cluster surveys, health survey, MIS, DHS, sub-Saharan Africa [14], logistic regression, multilevel regression, multinomial logistic, random effects, hierarchical, fixed effects, Anemi*, Anaemi*, Anaemia, Haemoglobin, Iron deficiency anaemia. These were the various terms used with appropriate Boolean connectors, “AND/OR”, in this order: (demographic health survey or AIDS indicator survey or malaria Indicator survey or multiple indicator cluster survey or health survey or MIS) and (sub-Saharan Africa) and (logistic regression OR multilevel regression OR multinomial logistic OR random-effects OR hierarchical OR fixed effects) AND (Anemi* OR Anaemi* OR Anaemia OR Haemoglobin OR Iron deficiency anaemia). We applied age-specific filters: 0–23 months and 24–59 months and year of publication from 1990 to 2020. The search was carried out between 22 to 26 June 2020

### 2.5. Selection Process

The reviewer, P.E.O., screened all the selected literature’s titles and abstracts as a function of the inclusion and extraction criteria. The reviewer conducted a full-text report screening and showed the reasons for excluding any paper. Two overseers of the project vetted this process. In the event of any contrary opinions, the most senior member of the overseeing team brokered any discrepancies noticed at this point.

### 2.6. Data Charting Management

The data extracted from the studies was first transferred into a Microsoft Excel (Microsoft Corporation, Washington, DC, USA) spreadsheet designed by the reviewer for this review. Relevant information from each study was included, such as the study outcome (anaemia status), various predictor variables concerning the child-related variables, parental/household-related variables and contextual or community-related variables, as well as the magnitude of their significant effects. Other items extracted were the titles of the studies and their authors/year of publication, the survey types, the number of samples or sample sizes (under five years) and the country in which the study was conducted.

## 3. Results

The results section reports the profile of the quantitative analysis of the risk factors associated with anaemia in under-five children in SSA. The results extracted are those reported from the studies that investigated the risk factors (both protective and harmful effects) that were evident from the Odds Ratios (OR), Relative Risk Ratios (RRR) or Risk Difference (RD) and Regression Estimates (RE).

### 3.1. The Study Profile Counts

A total of 217 studies (publications) were extracted from the electronic databases (Pubmed = 140, Scopus = 13 and Medline = 1) and another 63 studies from Google Scholar (using adjusted search terms to accommodate the length required for search terms in Google Scholar). Other sources searched include MeasureDHS journal publications, while using broad conditions such as “Anaemia” filtered for African countries and publication years between 1990 and 2020. The search yielded 43 studies. Twenty-six (26) other reviews were added from checking the references of the included studies and AJOL = 1. After the removal of 20 duplicate studies, the first scoping glance at the titles and abstracts eliminated a further 215 studies (publications). Fifty-two (52) studies were subjected to full-text examination, which resulted in retaining 20 publications for this review, after excluding 32 other studies (see Figure 1). The reasons for the exclusion of most of the studies were as follows:(i)Using other analytical methods rather than classical regression analysis.(ii)Studies did not report separate results for children 0–59 months.(iii)Studies that considered anaemia outcomes based on the maternal or child’s specifics.(iv)Studies not from SSA countries.(v)Papers without the full text available.

The 20 publications that met the inclusion criteria were further subjected to full-text scrutiny to answer the scoping review question. However, in this scoping review, the unit of analysis was the country for which a unique analysis of a unique data set was done. What this means is that a single study of two nations, in which the two countries were analysed separately according to their nationally representative surveys from each country, was counted as two studies. On the other hand, studies in several countries with pooled data as single analysis were counted as one study. Overall, a total of 24 countries’ studies were included in this review.

### 3.2. Characteristics of the Included Studies

Table 1 describes the profiles of the studies included in this scoping review. It shows the authors’ names and year of publication, the country of research, the title of the studies, survey type (i.e., the nationally representative survey type), the number of children of under five years used in each research and the classical statistical regression methods applied to determine the predictors of under-five anaemia in SSA countries. There is no preference in the order in which the authors are presented in Table 1.

The number of participants in each of the studies ranged from 933 under-five children captured in the study of Cape Verde by Semedo et al. [35] and published in 2014, to 96,804 under-five children from a pooled sample surveys of 27 SSA countries conducted by Moschovis et al. [23] and published in 2018. The statistical methods used to identify the potential risk factors for anaemia used in the selected studies include multivariate linear regression (two studies), multivariate binary logistic regression (nine studies), proportional regression (two studies), multilevel regression analysis (five studies), generalized linear mixed regression (5 studies), ordered logistic regression (1 study) and multinomial regression analysis (1 study). The earliest surveys considered in this scoping review were conducted in Ethiopia (EDHS), Malawi (MNCS) and Tanzania (TDHS) in 2016, while the oldest studies were BDHS (2001) and MDHS (2001), conducted in Benin and Mali, respectively.

### 3.3. The Study Profiles by Countries

The country of study is the unit of our review. Table 2 gives a breakdown of the study profiles by country—a total of twenty-four (24) unique country-based studies where examined from 20 extracted publications. A study conducted by Ngnie-Teta et al. [25] focused on two separate countries, Benin and Mali, and Ntenda et al. [36] focused on four different countries, Malawi, Mozambique, Namibia and Zimbabwe. Each of these countries used the country’s national representative survey, and their findings were also reported separately; hence, in this review, we also counted each as a separate study. However, the highest number of publications came from Ghana, Malawi and Ethiopia, with four studies each (representing 16% each). Uganda closely followed these with two studies (representing 8%).

The remaining countries had one study each. It is worth noting that one of the studies was classified as a multi-country study because it comprised 27 SSA countries, with the data pooled together and analysed as one study [23]. For Nigeria, there was no study that used classical regression analysis to determine the risk factors associated with under-five anaemia. The same was true for some other SSA countries.

### 3.4. Classification by Survey Types

Table 3 describes the various survey types captured in the review. A total of 27 nationally representative surveys were used in the 24 countries’ unique studies. Five different types of national representative sampled surveys were extracted, DHS, NHS, MIS, MNS and MICS. The most frequently used survey is the Demographic and Health Survey (DHS), with 19 studies (representing 70%), followed by the Malaria Indicator Survey (MIS) with four studies, and then the Multiple Indicator and Cluster Survey (MICS) with two studies (7%).

Other surveys captured in the review were the Micro-Nutrient Survey (MNS) and the National Health Survey (NHS), having one study each. One unique feature about these surveys is that they were conducted under the same technical assistance from ICF International through the MeasureDHS program, with the exemption of the NHS held in Cape Verde [35]. Some of these studies used a combination of two or more of these survey types.

### 3.5. Classification by Analytical Methods

Another essential feature of this scoping review is the identification of the varied analytical methods used to establish the risk factors associated with the anaemia status of children under five years of age in SSA. This is represented in Table 4.

Only studies that applied classical regression analysis tools were included. Nine (9) studies out of the 25 unique methods in the studies (36%) used multivariate binary logistic regression. Multilevel (which provides for hierarchical, random and fixed effects) and generalized linear mixed regression models were each applied in five studies. There was only one study that used multinomial regression analysis [21]. There were a couple of studies that used a combination of two or more of these analytical tools [18,23].

### 3.6. Classification of the Risk Factors

This section reports the distributions of the studies according to the variable groups, namely, the child-related, parental/caregiver-related, household-related and community-related variables.

#### 3.6.1. Child-Related Variables

Table 5 describes the distribution of the child-related variables in the included studies. Out of the twenty-four (24) unique, country-based studies, only one study [20] did not consider the age of the child as part of the child-related variables that were investigated. This implies that 96% of the studies evaluated considered the age of the child (0–59 months) as a risk factor, either as classified into different degrees of age groups or used as an interval variable. These studies found that the age of the child is a significant predictor of the development of anaemia among children under five years of age in SSA. The chances of having anaemia are much higher for children at a lower age (below 24 months) than at an older age [18,24,25,36]. For instance, Nambiema et al. [24] found a reduced effect of a child’s age with the Odds Ratio (OR) (OR = 0.22, 95% CI = 0.17–0.29); Ngnie-Teta et al. (Benin Republic study) [25] found an increased risk of developing anaemia for children aged 6–11 months (OR = 4.05, 95% CI = 2.40–7.09) and 12–35 months (OR = 2.81 95% CI = 1.99–4.52) when compared with children aged above 35 months. In another Malian study, Ngnie-Teta et al. [25] found that a child aged 6–11 months (OR = 1.73, 95% CI = 1.32–2.92) or 12–35 months (OR = 2.90, 95% CI = 2.24–3.92) is more likely to be anaemic when compared with a child older than 35 months.

Furthermore, the sex of the child as a risk factor predicting the chance of developing anaemia among children of under five years in SSA was reported in 17 studies (representing 71%). Almost all of these 17 studies reported significant variations in the status of anaemia by sex. In almost all the studies that reported sex as a risk factor, it was found that a male child was more prone to having anaemia than a female child [18,22,23,26,29,32].

Comorbidities of anaemia with having diarrhoea and fever (in the last two weeks before the survey) were reported in 12 (50%) and 11 (46%) studies, respectively. Moschovis et al. [23] reported a slight harmful effect of anaemia for a child who had non-bloody diarrhoea (OR = 1.11, 95% CI = 1.04–1.18) or bloody diarrhoea (OR = 1.21, 95% CI = 1.07–1.36) when compared with a child without diarrhoea in the last two week before the survey [23]. However, Jones et al. [20] found no significant effect (OR = 1.1, 95% CI = 0.77–1.6). Significantly higher odds of developing anaemia among children of under five years in SSA was also reported for children that had a fever in the last two weeks before the survey than those that had not (OR = 1.42, 95% CI = 1.36–1.49, [23]; and OR = 1.46, 95% CI = 1.04–2.32) in Mali [25]. Besides the strong relationship between the anaemia and nutrition indicators, stunting as a risk factor was examined in nine (36%) of the included studies. In comparison, “wasting” was examined in three (12%) of the included studies. Moreover, nutrition status (a composite of all the nutrition indicators) was only reported in one (4%) of the 24 studies [24]. The odds of having under-five anaemia was 1.82 times higher for a malnourished child than a child who is well nourished [24].

Treatment for intestinal worms in the last 6 months was reported as a significant factor in Moschovis et al. (OR = 1.06, 95% CI = 1.02–1.11) [23], but not significant in Jones et al. (OR = 0.98, 95% CI = 0.76–1.3) [20]. Birth order as a risk factor for anaemia in children of under five years of age was reported in six (6) countries studies. Two studies reported significant harmful effects, but contrary to one another. Mischovis et al. [23] found that having a lower birth order is significantly harmful in developing under-five anaemia compared with having more than three birth orders, while Ngnie-Teta et al. (Benin Republic study) [25] concluded that being born as the sixth birth order or later is significantly two-folds more harmful than a single birth order (OR = 2.05, 95% CI = 1.02–3.97).

#### 3.6.2. Distributions of Parental/Caregivers-Related Variables

Mother’s age, work status, educational status and anaemia status was frequently reported in the included studies. Table 6 indicates that the mother’s educational status was reported in 21 (84%) of the studies, followed by mother’s age (13 studies) and mother’s anaemia status (12 studies). Therefore, among the parental/caregiver related variables, 84% of the studies placed the mother’s educational status as one of the most frequently considered risk factors of anaemia in under-five children in SSA. The results from most of these 21 studies showed that, as the level of educational status of the mother increases, the chance that the child will develop anaemia decreases. For instance, Nambiema et al. [24] found that a child whose mother has a secondary level of education and above has a lower adjusted odds of developing anaemia than a child whose mother has no education (OR = 0.67, 95% CI = 0.52–0.86). There was also a clear-cut pattern of how the variation in the mother’s age affected the chances a child developing anaemia. For instance, Moschovis et al. [23], Asresie et al. [34] and Ojoniyi et al. [32] reported a drop in the odds of having anaemia among children of under five years as the mothers’ age increases.

The mother’s anaemia status was reported in 12 (50%) of the studies included in this scoping review. Moschovis et al. [23] found that a child whose mother was anaemic had an 85% greater odds of having anaemia than another child whose mother was not anaemic (OR = 1.85, 95% CI=1.76–1.95). Iron supplementation during pregnancy was reported in only one study [21] and was not a significant risk factor (RRR = 1.00, 95% CI = 0.7–1.6).

#### 3.6.3. Distributions of Household-Related Variables

Another critical component of the risk factors associated with anaemia among children under five years in SSA was the household-related variables. Table 7 shows the details of the distribution of various household-related risk factors. Wealth status, a proxy of household socioeconomic status, was one among many factors that drew more attention in this category of risk factors. Twenty-one (21) studies, representing 87%, considered for this scoping review were examined for wealth status. Most of the studies that reported significant effects of household wealth status on under-five anaemia in SSA countries established that the higher the wealth quintiles, the lower the risk of developing anaemia among under-fives [19,23,29,32]. The Hershey et al. [19], Mohammed et al. [22] and Moschovis et al. [23] studies found, respectively, that being in the richest category (OR = 0.55, 95% CI = 0.44–0.70; OR = 0.48, 95% CI = 0.33–0.63; and OR = 0.417, 95% CI = 0.287–0.547, respectively) had a significant protective effect against under-five anaemia. On the contrary, Ntenda et al. [27] found in their Malawi study (OR = 0.81 95% CI = 0.60–1.08), Mozambique study (OR = 0.48, 95% CI = 0.38–1.24) and Namibia study (OR = 0.76, 95% CI = 0.53–1.11) that being in higher quintiles of wealth status is a protective but not a significant factor.

Closely following the effect of wealth status in this category of household-related risk factors was the place of residence (that is, whether the household under study is in a rural or urban area). With 18 (75%) studies, the place of residence was the second most examined household-related variable as a risk factor associated with anaemia among children of under five years in SSA countries. Among the studies that reported a significant association of place of residence, there was no clear-cut conclusions relating to the comparison of rural and urban dwellers. For instance, Menon and Yoon (OR = 0.768, 95% CI = 0.592–0.996) [30], Mohammed et al. [22] and Moschovis et al. [23] reported a protective effect for rural compared to urban areas, while Ngnie-Teta et al. [25], in their Malian study (OR = 2.04, 95% CI = 1.38–3.44), as well as Nambiema et al. (OR = 0.66, 95% CI = 0.53–0.82) [24] and Dwumoh et al. (OR = 0.53, 95% CI = 0.46–0.65) [18] found that it was more likely for a child in the rural area to develop anaemia than in the urban area of SSA. In turn, Ntenda et al. [26] discovered it was more harmful being in a rural than urban area, but it was not a significant factor (OR = 1.27, 95% CI = 0.53–3.01).

Other risk factors that were of utmost importance in many of the studies included in this scoping review include the following:(i)Household size, in four (17%) studies.(ii)The number of children that were under-five years living in the same household (17%).(iii)Having an improved source of drinking water, reported in eight (33%) studies.(iv)The child slept under a mosquito net the previous night before the survey (17%).

Two studies reported findings on the use of biomass for cooking in three country-related studies. Contrary to the expectation from other studies not included in this review [37,38], in that exposure to biofuel for cooking and heating may result in harmful effects, with likely developing anaemia compared to those children exposed to cleaner cooking and heating fuel, the studies [23,36] included in this scoping review found the opposite conclusion. Moschovis et al. (OR = 0.99, 95% CI = 0.90–1.10) [23], Ntenda et al. [27], in Mozambique study (OR = 0.93, 95% CI = 0.50–1.73) and in a Namibia study (OR = 0.92, 95% CI = 0.58–1.45), reported a protective association regarding the use of biofuel for cooking, but it was not significant.

#### 3.6.4. Distribution of Study Characteristics by Community-Related Variables

Community-based risk factors (Table 8) were not very popular in all the studies added to this review. The few that are of general importance are the community poverty and wealth statuses (these were computed as the mean per cent of the community wealthiest households), and community female educational status (computed as the mean percent of women in the community that has primary education and above). There were four studies in this category. Other variables included the distance to the nearest health facility and level of access to safe drinking water for the community.

Some of the studies generally included their countries’ regions or place of residence as community risk factors. Since the regions were not unique for all studies, we dropped them from the list of risk factors at the community levels.

## 4. Discussion

The aim of this scoping review was to identify and evaluate the studies that performed classical regression analysis on nationally representative health survey data to identify the individual socioeconomic, demographic and contextual risk factors associated with developing anaemia among children under five years of age in sub-Sahara Africa (SSA). The review identified 20 studies and the risk factors for anaemia were classified as child-related, parental/household-related and community or area-related factors. The risk factors for anaemia identified included age, birth order, sex, comorbidities (such as fever and diarrhoea), malnutrition or stunting, maternal education, maternal age, mother’s anaemia status, household wealth and place of residence.

This review describes the existing pieces of evidence about results obtained in different studies using nationally representative samples in sub-Saharan Africa countries. The broad scope elicited information from studies using a range of classical regression methods, study designs and risk factors associated with anaemia among children of under five years in SSA. We have provided from the onset a broad research question that guided the review. We screened some electronic databases, search engines and grey literature-bases to draw out some substantial scholarly works that have been formally published. The comprehensive search, which lasted for a week, yielded a reasonable number of pieces of literature after applying some inclusion and exclusion criteria. The data charting form designed for this study extracted the relevant information, ranging from the authors’ name and year of publication, the study design, the topics, the analytical techniques, the numbers in the sample and the identified risk factors (classified under child-related, parental/caregiver-related, household-related and community-related) associated with under-five anaemia in SSA. Out of a total of forty-six (46) SSA countries [39], this scoping review could extract publications involving only fifteen (15) countries (representing 32%), but there were twenty-four (24) unique country studies, which means some countries recorded more than one study.

The results from this study showed that the overall prevalence of under-five anaemia from SSA countries is very high, ranging from 29% in Malawi from a study by Ntenda et al. [36] to 83% in Mali from a study conducted by Ngnie-Teta et al. [25]. Going by the WHO’s classification of anaemia status, wherein severe anaemia is a prevalence of more than 40%, then most SSA countries could be classified as highly burdened with severe under-five anaemia. These findings agree with the results found among children in India (which was found to be higher than 50%) [40], Nepal (46.4%) and Pakistan (62.5%) [41]. The findings from almost all the studies that considered gender as a risk factor also concluded that male under-five children are more prone to anaemia than their female counterparts. These findings do not agree with the study conducted among children in Kuwait [42], which found that female children between the age of 6 months and 3 years have a higher prevalence rate of anaemia than the male children in the same age bracket.

Although in SSA, anaemia in under-fives has become one of the severe public health burdens for most countries’ health sectors, little has been done in conducting large-scale research for informed decision-making. Nigeria is one of the countries in SSA that has increasingly a very high prevalence of under-five anaemia, from 60% in 2015 to 68% in 2018 [5,6]. Though there are already three different nationally representative surveys [5,43,44] conducted with data on anaemia status, surprisingly from our search strategies (both inclusion and exclusion criteria) there was no single study on risk factors for anaemia among children aged under five years that used classical regression analysis methods to analyse the nationally representative data set found. This is a gap in knowledge that this is yearning to be addressed.

Most of the studies in this scoping review applied classical regression methods (multivariate linear regression and logistics) that only evaluate the risk factors at the individual and household levels, without accounting for the area or contextual risk factors. This lack of consideration for area variables often leads to random effect errors (heterogeneity). With less than 20% of the included studies using multilevel analysis techniques, most of the studies have neglected these critical components of determinants of anaemia status among children under five years in SSA.

One of the challenges in addressing anaemia among children using the best nutritional intervention is the utmost attention given to IDA as a proxy for nutritional anaemia at the expense of other micronutrient deficiencies. Most of the studies considered in this review used IDA to determine the anaemia status from the measure of haemoglobin level. Though most of the nationally representative survey collected data on some other micronutrient deficiencies, such as folate, vitamin A and B12, these are seldom considered as proxies for anaemia status in children.

Moreover, the alarming prevalence of anaemia among children under five years in most SSA countries calls for possible interventions through making iron-fortified food available to infants beyond the period of breastfeeding. Cultivation of iron-fortified crops or organic and healthy food production by these SSA countries is a way to address these deficiencies. Pregnant women as a matter of policy practice will require a substantial number of recommended doses of iron-enriched foods and supplements so that their infants are born endowed with enough iron reserves to sustain them through the nursing period.

Finally, the findings in most of the studies in this review showed that anaemia in children under five years in SSA is associated with other common but fatal childhood illnesses (diarrhoea, fever and malnutrition). Comorbidities and multi-morbidity are typical health issues that in the past years have been associated with adults over 60 years [45]. However, in the last few decades, it has increasingly becoming associated with children [46,47]. How this can affect a health system, particularly in SSA countries, is an emerging area for study [48]. Determinants of co-existence of illness are better examined with multinomial analysis while using classical regression methods. In this scoping review, only one study used multinomial regression analysis [21]. Results from any research that investigate the overlapping associations in comorbidities and multi-morbidity would be exciting although it may require some methodological rigours. However, they are worth investigating.

## 5. Strengths and Limitations

In our view, this is likely to be the first scoping review to provide information concerning the risk factors associated with anaemia in children under five years of age using nationally representative surveys from sub-Saharan Africa countries. The strength of the evaluation is the rigorous checks from the teams involved from different institutions and the outcomes that can be a pointer to a grey area of gaps in the study that needs urgent attention. For instance, the study has revealed some SSA countries where under-five anaemia studies that use classical regression analysis on nationally representative surveys are lacking. Most of these countries have a very high prevalence of under-five anaemia

It is acknowledged that there are a several limitations in this study. The numbers of studies meeting the inclusion criteria were very few, perhaps because of the inclusion/exclusion criteria. It is also possible not to have identified all the papers in this area of research in view that we only included studies written in English [14]. However, researchers from developing countries have not shown much zeal in areas that needed more statistical rigour to analyse nationally representative surveys, such as the ones considered in this review. We are confident that we may have identified significant relevant texts meeting the inclusion/exclusion criteria. However, it is possible that we have omitted some insignificant risk factors during the charting and extraction of the information. We also recognized that we neither carried out any publication bias assessment nor did we evaluate the quality of the studies included due to the scoping review design [48].

## 6. Conclusions

A considerable amount of resources are spent annually to conduct nationally representative surveys across over 90 countries, but studies conducted to take sufficient advantage of these data sets that will influence practice and policy are too few to justify the huge expenditures [14]. This may have been one of the reasons why little or no significant achievements have been made in curbing the harm caused by these diseases in SSA. Therefore, research is urgently needed to analyse the vast nationally representative datasets for informed decision-making, to tackle the numerous public health issues in SSA. This is, however, without prejudice of the need to conduct studies focusing on specific diseases. Interestingly, our review found no studies on the risk factors for children under five years in Nigeria that used classical regression analysis methods that met the inclusion and exclusion criteria.

## Figures and Tables

**Figure 1 ijerph-17-08829-f001:**
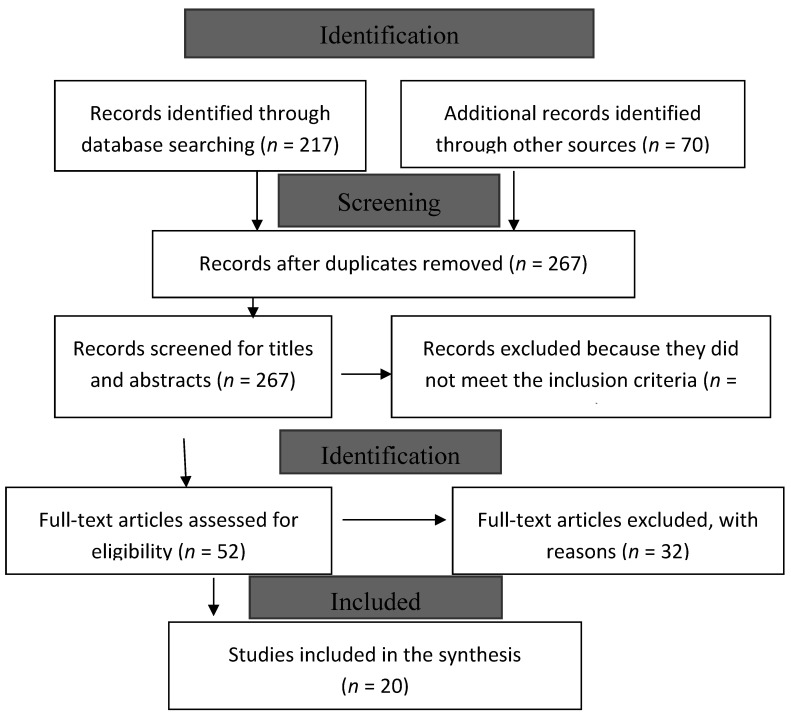
Flowchart showing the inclusion process.

**Table 1 ijerph-17-08829-t001:** Characteristics of the selected studies on anaemia (*n* = 24).

Author(s) (Year)	Country	Title of Study	Survey Type	Prevalence of Anaemia	Participation	Methods
Dwumoh et al. (2014) [18]	Ghana	Determinant of factors associated with child health outcomes and service utilization in Ghana: Multiple indicator cluster survey conducted in 2011	MICS	There was no % prevalence reported	7550	Binary logistic regression models and multiple linear regression
Hershey et al. (2017) [19]	Malawi	Malaria Control Interventions Contributed to Declines in Malaria Parasitaemia, Severe Anaemia, and All-Cause Mortality in Children Less Than 5 Years of Age in Malawi, 2000–2010	DHS, MICS and MIS	Prevalence of severe anaemia in 2010 was 8.7%	Proportion	Multivariable, random effects logistic regression models
Jones et al. (2018) [20]	Ghana	Livestock ownership is associated with higher odds of anaemia among preschool-aged children, but not women of reproductive age in Ghana	DHS	Moderate anaemia was 56.4%, mild anaemia was 40.2%	2735	Multiple binary logistic regression models
Machisa et al. (2013) [21]	Swaziland	Biomass fuel use for household cooking in Swaziland: is there an association with anaemia and stunting in children aged 6–36 months?	DHS	51.8% in children 6–36 months	1150	Multinomial logistic regression analyses
Mohammed et al. (2019) [22]	Ethiopia	Household, maternal, and child-related determinants of haemoglobin levels of Ethiopian children: hierarchical regression analysis	DHS 2016	71.92% in the study population (6–23 months)	2902	Hierarchical linear regression analysis
Moschovis et al. (2018) [23]	27 SSA countries	Individual, maternal and household risk factors for anaemia among young children in sub-Saharan Africa: a cross-sectional study	DHS 2008–2014	59.9% among children 6–59 months	96,804	Multiple linear regression or multiple binary logistic regression
Nambiema et al. (2019) [24]	Togo	Prevalence and risk factors of anaemia in children aged from 6 to 59 months in Togo: analysis from Togo demographic and health survey data	DHS 2013–2014	70.9% among children 6–59 months	2890	Logistic regression models
Ngnie-Teta et al. (2007)[25]	Benin	Risk factors for moderate to severe anaemia among children in Benin and Mali: insights from a multilevel analysis	DHS 2001	82%	2284	Multilevel binary logistic model
	Mali	Risk factors for moderate to severe anaemia among children in Benin and Mali: insights from a multilevel analysis	DHS 2001	83%	2826	Multilevel binary logistic model
Ntenda et al. (2019) [26]	Malawi	Clinical malaria and the potential risk of anaemia among preschool-aged children: a population-based study of the 2015–2016 Malawi micronutrient survey	2015–2016 MNS	29%	1051	Multivariate binary logistic regression models
Ntenda et al. (2018) [27]	Malawi	Multilevel Analysis of the Effects of Individual- and Community-Level Factors on Childhood Anaemia, Severe Anaemia, and Haemoglobin Concentration in Malawi	2010 DHS	63%	2597	Multilevel linear regression models
Kawo et al. (2018) [2]	Ethiopia	Multilevel Analysis of Determinants of Anaemia Prevalence among Children Aged 6–59 Months in Ethiopia: Classical and Bayesian Approaches	2010 DHS	42.8%	5507	Multilevel binary logistic regression analysis
Immurana and Arabi (2017) [28]	Ghana	Socioeconomic factors and child health status in Ghana	2014 DHS	71.11% male and 67.95% female children	2220	Binary probit model
Candia (2017) [29]	Uganda	Influence of malaria on anaemia levels among children less than 60 months of age	MIS	53.22%	4940	Ordered logistic regression model
Menon and Yoon (2015) [30]	Uganda	Prevalence and Factors Associated with Anaemia among Children Under 5 Years of Age—Uganda, 2009	2009 MIS	60% of children under five years	4065	Multivariate binary logistic regression model
Nikol and Anthamatten (2013) [31]	Ghana	Childhood anaemia in Ghana: an examination of associated socioeconomic and health factors	2008 DHS	79.8%	2055	Generalized linear mixed regression model
Ojoniyi et al. (2019) [32]	Tanzania	Does education offset the effect of maternal disadvantage on childhood anaemia in Tanzania? Evidence from a nationally representative cross-sectional study	2015–2016 DHS/MIS	58.6%	7916	Proportional odds model
Muchie (2016) [33]	Ethiopia	Determinants of severity levels of anaemia among children aged 6–59 months in Ethiopia: further analysis of the 2011 Ethiopian demographic and health survey	2011 DHS	28.6% were severely/moderately anaemic and 21.7% were mildly anaemic	7636	Proportional odds model of ordinal logistic regression
Asresie et al. (2020) [34]	Ethiopia	Determinants of Anaemia among Children Aged 6–59 Months in Ethiopia: Further Analysis of the 2016 Ethiopian Demographic Health Survey	2016 DHS	58% of children 6–59 months	8462	Binary Logistic regression analyses
Semedo et al. (2014) [35]	Cape Verde	Prevalence of anaemia and associated factors among children below five years of age in Cape Verde, West Africa	NHS	51.8%	933	Hierarchical model for multiple analysis
Ntenda et al. (2018) [36]	Malawi	Maternal anaemia is a potential risk factor for anaemia in children aged 6–59 months in Southern Africa: a multilevel analysis	2010 DHS	63.8%	2507	Generalized linear mixed models (GLMMs)
	Mozambique	Maternal anaemia is a potential risk factor for anaemia in children aged 6–59 months in Southern Africa: a multilevel analysis	2013 DHS	70%	1933	Generalized linear mixed models (GLMMs)
	Namibia	Maternal anaemia is a potential risk factor for anaemia in children aged 6–59 months in Southern Africa: a multilevel analysis	2013 DHS	49%	1116	Generalized linear mixed models (GLMMs)
	Zimbabwe	Maternal anaemia is a potential risk factor for anaemia in children aged 6–59 months in Southern Africa: a multilevel analysis	2010–2011 DHS	58.6%	2578	Generalized linear mixed models (GLMMs)

Note: Multiple Indicators Cluster Survey (MICS); Demographic and Health Survey (DHS); Malaria Indicator Survey (MIS); National Household Survey (NHS); Micronutrient Survey (MNS).

**Table 2 ijerph-17-08829-t002:** Study profiles by country.

Country Specific Articles	Number	%	References
Ghana	4	16.8	[18,20,28,31]
Ethiopia	4	16.8	[2,22,33,34]
Mali	1	4.2	[25]
Benin	1	4.2	[25]
Uganda	2	8.4	[29,30]
Tanzania	1	4.2	[32]
Malawi	4	16.8	[19,26,27,36]
Swaziland	1	4.2	[21]
Multi-country	1	4.2	[23]
Togo	1	4.2	[24]
Cape Verde	1	4.2	[35]
Mozambique	1	4.2	[36]
Namibia	1	4.2	[36]
Zimbabwe	1	4.2	[36]
	24 *	100	

* A total of twenty-four (24) unique country-based studies were examined from 20 extracted studies (publications).

**Table 3 ijerph-17-08829-t003:** Description of the survey types in this anaemia review.

Survey Type Specific	*N*	%
Demographic and Health Survey (DHS)	19	70
Multiple Indicator Survey (MIS)	4	15
Micronutrient Survey (MNS)	1	4
Multiple Indicator Cluster Survey (MICS)	2	7
National Health Survey (NHS)	1	4
Total	27 *	100

* Some studies used more than one survey (see Table 1).

**Table 4 ijerph-17-08829-t004:** Classification of the analytical methods.

Analytical Methods	*N*	%	References
Multivariate Linear Regression	2	8	[18,23]
Multivariate Logistic Regression	9	36	[18,19,20,23,24,26,28,30,34]
Proportional Ordinal Logistic Regression	3	12	[29,32,33]
Multilevel Regression	5	20	[2,22,25,27]
Generalised Linear Mixed Regression Model	5	20	[31,36]
Multinomial Regression	1	4	[21]
Total	25 *	100	

* Some studies used more than one analysis technique.

**Table 5 ijerph-17-08829-t005:** Distribution of the child-related variables for anaemia from the 24 country-specific results.

Risk Factor: Child-Related Variables	Number of Studies Which Investigated the Risk Factor (%)	References
Age of the child	23/24 (96%)	[2,18,19,21,22,23,24,25,27,28,29,30,32,33,34,35,36]
Sex of the child	17/24 (71%)	[2,18,19,20,21,22,23,24,25,26,27,28,29,30,31,32,35,36]
Has health insurance	4/24 (17%)	[18,28,31,32]
Perceived birth size	3/24 (12%)	[2,22,33]
Ever had vaccination status	1/24 (4%)	[35]
Product of multiple births	2/24 (8%)	[23,32]
Preceding birth interval	1/24 (4%)	[23]
Birth order	6/24 (25%)	[23,25,26,28,33]
Iron supplement	4/24 (17%)	[2,20,21]
Duration of breastfeeding	4/24 (17%)	[20,21,22,35]
Breastfeeding	2/24 (8%)	[22,23]
Had diarrhoea in last 2 weeks	12/24 (50%)	[20,21,23,25,26,34,35,36]
Had fever in last 2 weeks	11/24 46%)	[20,21,23,26,31,35,36]
Vitamin A consumption	4/24 (16.6%)	[20,21,22,27]
Min Dietary Diversity (MDD)	1/24 (4%)	[22]
Min Meal Frequency (MMF)	1/24 (4%)	[22]
Treatment for intestinal worms in the last 6 months	3/24 (12%)	[20,23,36]
Nutrition status	1/24 (4%)	[24]
Stunting	9/24 (37%)	[2,23,25,27,36]
Wasting	3/24 (12%)	[2,27]
Underweight	5/24 (20%)	[36]
Overweight	1/24 (4%)	[32]
Malaria status (blood smear)	3/24 (12%)	[19,24,26]
Malaria status (rapid test)	1/24 (4%)	[30]

**Table 6 ijerph-17-08829-t006:** Distribution of the study characteristics of the parental/caregiver-related variables for anaemia.

Parental/Caregiver-Related Variables	Number of Studies That Investigated the Risk Factors	References
Mother’s age in years (grouped)	13/24 (54%)	[18,22,23,25,26,27,28,32,33,34,36]
Mother’s age at child’s birth	1/24 (4%)	[21]
Mother working Status	6/24 (25%)	[2,24,28,32,33,34]
Mother’s educational status	20/24 (83%)	[2,18,20,21,22,24,25,27,28,29,30,31,32,33,34,36]
Father’s educational status	4/24 (17%)	[25,28,33]
Father is alive at the date of the survey	1/24 (4%)	[24]
Mother’s marital status	3/24 (12%)	[2,28,32]
Mother’s body mass index (kg/m^2^)	4/24 (17%)	[21,22,23,31]
Mother’s anaemia status	12/24 (50%)	[21,22,23,24,26,27,31,33,34,36]
ANC attendance	1/24 (4%)	[22]
Religion status	2/24 (8%)	[28,33]
Mother’s iron supplementation during pregnancy	¼ (4%)	[21]

**Table 7 ijerph-17-08829-t007:** Distribution of study characteristics by household-related variables.

Household-Related Variables	Number of Studies Which Investigated the Risk Factor	References
Wealth status	21/24 (87%)	[2,18,19,20,21,22,23,24,25,27,28,29,30,31,32,34,36]
Place of residence	18/24 (75%)	[2,18,20,22,23,24,25,26,27,29,30,36]
Household had bed net	2/24 (8%)	[20,30]
Age of household head	1/24 (4%)	[28]
Recent anti-malaria indoor residual spraying of household	1/24 (4%)	[20]
Household size	4/24 (17%)	[21,23,25,34]
Number of children under 5 in the household	3/24 (12%)	[2,32,33,34]
Water source outside the premises	1/24 (4%)	[23]
Improved source of drinking water	8/24 (33%)	[2,20,22,23,25,29,33]
Improved type of toilet facilities	2/24 (8%)	[20,23]
Unsafe stool disposal	1/24 (4%)	[23]
Improved floor material type	1/24 (4%)	[23]
Sex of household head	2/24 (8%)	[20]
Shared toilet facilities with other household members	1/24 (4%)	[23]
Use biomass for cooking	3/24 (12%)	[23,36]
Under-fives slept under mosquito nets last night	4/24 (17%)	[19,21,25]
Household ownership of livestock	1/24 (4%)	[20]

**Table 8 ijerph-17-08829-t008:** Distribution of the study characteristics by community-related variables.

Community Variables	Number of Studies Which Investigated the Risk Factor	References
Community wealth	4/24 (17%)	[20,36]
Community female education	4/24 (17%)	[27,36]
Community distance to health facility	3/24 (12%)	[36]
Community safe water access	3/24 (12%)	[36]

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
