# Peer review of "A Scoping Review of the Risk Factors Associated with Anaemia among Children Under Five Years in Sub-Saharan African Countries"

_ijerph, 2020, doi:10.3390/ijerph17238829_

Round 1

Reviewer 1 Report

As is common in Scoping reviews, the goal was not necessarily to make new discoveries, but to characterize the state of the field and the extent of the available literature.  This summary does a good job of accomplishing that goal.  It is scholarly and encyclopedic.

The predictive variables are already known, but it is still interesting to see them summarized with respect to how many studies considered them.

If they want to address one more point in Discussion, I might have considered the challenges of distinguishing a single micronutrient deficiency from undernutrition/malnutrition in general.  That becomes important when trying to decide on the best nutritional interventions, as well as when trying to attribute health consequences to just iron deficiency.

They could also discuss the challenge of delineating iron deficiency anemia from the anemia of chronic disease.  In particular, in regions where there are parasitic infections, including intestinal worms, as well as malaria, the nutritional deficiencies typically co-exist with infections.   his point is relevant in part to the association they discuss between IDA and diarrheal disease in many young children.  The question in part is which came first?   The nutritional deficiencies or the illness.

Finally, one wonders if there should be any mention in the Discussion of possible interventions, including the increased use of iron-fortified crops.  Or where possible the earlier provision of foods that may have higher levels of iron for infants who are being exclusively breast-fed beyond 6 months of age.  Although clearly a major challenge at the policy level is ensuring that pregnant women are have access to sufficient iron so that their infant are born endowed with sufficient iron reserves to sustain them through the nursing period.  Breast milk alone does not have enough iron to meet the needs of a rapidly growing infant.  When there is a shortfall both during the prenatal period and then after birth, the progression to IDA is inevitable.   But it is not easy to increase the amount of iron in breast milk.

Writing

Line 35.    Probably can change ‘implies’ to ‘indicates’

Line 37.  Change ‘one that’ to ‘ones who’

Line 38.   Not sure that ‘however’ is needed at the start of the sentence

Line 61.  Probably a better word than ‘menace’ is needed.  Maybe something less dramatic, such as ‘the harmed caused by’

Line 74.  Not sure why AND OR were capitalized.

Line 88.   The word ‘it’ should be inserted after ‘but’.   Period needed at end of sentence

Line 143.   The t in Text does not need to be capitalized.

Line 160.  Change ‘at the event’ to ‘in the event’

Line 164   change ‘deposit’ to ‘transfer’

Line 173.   Awkward sentence that should be rewritten.   ‘results extracted were those…..”

Line 197.  Data set should be change to data obtained.

Line 249.    Awkward writing.  ‘corresponding depths of researches’.  The point is important with respect to whether the research community has taken sufficient advantage of the survey and epidemiological knowledge to influence practice and policy.

Line 252.   It is not common in English to write ‘researches’.  Better to substitute ‘studies’ or “investigations’

Line 254.  ‘no evidence of’ is not needed.  It can just be ‘no studies’.

Reviewer 2 Report

This paper presents the findings of a scoping review of the publish literature concerning risk factors for anaemia in under five-year olds in sub-Saharan Africa. Using the commonly reported literature databases with the associated limitations which the authors recognise, they conclude from the 20 relevant studies several factors.

While the authors have undertaken a useful piece of work which will add to the available literature on the topic, the paper requires some editing and a number of corrections to be made. For example on line 79 the wording in the brackets makes no sense and line 197 of the second section (bottom of page 16) requires completion. There are a number of other grammatical and spelling errors which need addressing.

That said the content of the paper itself is likely to add value to the understanding and hopefully reduction of the problem of anaemia and the authors need to be careful with the recommendation starting on line 248 as to the value of the studies undertaken and might wish to reword the sentence. One of the alternatives is of course to conduct studies into very specific health issues as opposed to national studies.
